# Photocatalytic oxidation of methane over silver decorated zinc oxide nanocatalysts

Xuxing Chen[1,2], Yunpeng Li[1], Xiaoyang Pan[1], David Cortie[3], Xintang Huang[2] & Zhiguo Yi[1]

The search for active catalysts that efficiently oxidize methane under ambient conditions remains a challenging task for both C1 utilization and atmospheric cleansing. Here, we show that when the particle size of zinc oxide is reduced down to the nanoscale, it exhibits high activity for methane oxidation under simulated sunlight illumination, and nano silver decoration further enhances the photo-activity via the surface plasmon resonance. The high quantum yield of 8% at wavelengths $< 400$ nm and over 0.1% at wavelengths $\sim 470$ nm achieved on the silver decorated zinc oxide nanostructures shows great promise for atmospheric methane oxidation. Moreover, the nano-particulate composites can efficiently photo-oxidize other small molecular hydrocarbons such as ethane, propane and ethylene, and in particular, can dehydrogenize methane to generate ethane, ethylene and so on. On the basis of the experimental results, a two-step photocatalytic reaction process is suggested to account for the methane photo-oxidation.

[1] Key Laboratory of Design and Assembly of Functional Nanostructures and Fujian Provincial Key Laboratory of Nanomaterials, Fujian Institute of Research on the Structure of Matter, Chinese Academy of Sciences, Fuzhou 350002, China. [2] Department of Physics, Institute of Nanoscience and Nanotechnology, Central China Normal University, Wuhan 430079, China. [3] Research School of Chemistry, The Australian National University, Canberra, Australian Capital Territory 2601, Australia. Correspondence and requests for materials should be addressed to Z.Y. (email: Zhiguo@fjirsm.ac.cn).

Methane, as the principal constituent of natural gas, is widely used as a fuel and is an important raw material in industrial chemical processes. In view of its utility for improving the quality of human life the emissions of methane were ignored as a trivial matter for a long time and this has led to a significant increase in the atmospheric methane concentration since the industrial revolution[1–3]. Nowadays, with the increasing concern about environmental pollution and climate change, the negative impact of methane emissions is attracting more attention[4–6]. In comparison with other greenhouse gases, methane is responsible for nearly one-fifth of anthropogenic global warming. Over the course of a century, it has a greenhouse gas effect that is more than twenty times greater than the effect from the equivalent mass of carbon dioxide[1,2,7]. More seriously, global warming and shale gas exploitation are likely to enhance methane release from a number of sources. Therefore, conversion of atmospheric $CH_4$ into equimolar amounts of $CO_2$ can have a significant impact on reducing global warming.

Given the high C–H bond energy ($434 kJ mol^{-1}$) and the non-polar nature of the $CH_4$ molecule, thermo-catalysis involving precious metals or transition metal oxides have been extensively studied during the past decades for the conversion of methane[8–14]. The high reaction temperature ($\sim 400\,°C$) and inefficiency in removing trace amounts of methane are drawbacks of this approach. Semiconductor photocatalysis, as a technology utilizing sunlight, has been shown to be promising in both water splitting and environmental remediation[15–19]. Earlier reports have also shown that by using the approach of photocatalysis, activation and oxidation of methane can take place even at room temperature at atmospheric pressure[20–23]. The efficiency of photocatalytic oxidation of methane, however, remains notoriously low even under light irradiation using ultraviolet sources.

In our preliminary studies, we fabricated a range of semiconductors including $SrTiO_3$, $KNbO_3$, CdS, $Cu_2O$, $BiVO_4$, $g-C_3N_4$ and $Ag_3PO_4$, and so on. that have shown strong capabilities to drive water cleavage under light irradiation, using solid state reaction, hydrothermal, or other modified methods to examine their performance on driving methane photo-oxidation. None of the aforementioned semiconductors, which are known to have strong reduction or oxidation capabilities, exhibit any activity for $CH_4$ photo-oxidation except P25 $TiO_2$ which shows a moderate photo-activity. Heterojunction interface design[24], morphology control[25] and band edge modulation[26] were successively also used to fabricate photoactive materials to address the photo-oxidation of small molecular hydrocarbons. Some small molecular hydrocarbons such as $C_2H_6$, $C_3H_8$ and $C_2H_4$ can be efficiently treated by these techniques, however, effective treatment of methane still remains a great challenge.

In light of the possibility that zinc ions may play an important role in methane activation[27], we then turned to zinc containing compounds such as ZnO to examine its activity on photo-oxidizing methane. It should be noted that, although it has been extensively studied, ZnO has never been recognized as an efficient photocatalyst because of its limited light-harvesting ability and serious photo-corrosion problem.

Efficient photocatalysts need to: (1) absorb sunlight across the ultraviolet–visible (UV–vis) region to produce electrons and holes; (2) separate the electrons and holes in space to prevent their recombination; (3) have suitable redox potentials to drive the photo-oxidative reactions. It is challenging to satisfying all the requirements in a single material. In particular, the generation of the active oxygen species $O_2^-$ and $^\bullet OH$ radicals is crucial step for the photocatalytic oxidation of hydrocarbon species, which means semiconductors with a conduction band minimum higher than the potential of $O_2/O_2^-$ ($-0.16 V$ versus NHE)[28] and valence band maximum lower than the $^\bullet OH/OH^-$ ($+2.59 V$ versus NHE)[24,29] potential are needed for organic pollutant degradation.

Ag decorated ZnO is chosen in this study not only because ZnO is an inexpensive semiconductor with large band gap satisfying the band edge potential requirement, but also because it fulfills the following materials design considerations (Fig. 1): (1) the polar structure renders fast separation and transportation of photo-generated electrons and holes[25]; (2) rich defective surfaces benefit surface reactions[30]; (3) nano silver decoration may function as both a co-catalyst and a light-harvesting medium[31]; (4) applying ZnO in gas phase photo-degradation may halt the photo-corrosion that constantly occurs in aqueous solutions[32,33]. The experimental results show that a nanoscale ZnO can efficiently oxidize methane under simulated sunlight irradiation and nano silver decoration further improves the activity to a high level even under visible light illumination.

## Results

**Characterization of Ag–ZnO nanocatalysts.** The as-prepared ZnO and Ag–ZnO nanopowders have a Brunauer–Emmett–Teller (BET) surface area of 45.9 and 40.2 $m^2 g^{-1}$, respectively. X-ray diffraction (XRD) analysis identified the hexagonal wurtzite structure type of ZnO (JCPDS file no. 99-0111) for all samples and no diffraction peaks were detected for Ag owing to its low volume fraction (Fig. 2a) and fine particle size (to be shown in Fig. 2e). The UV–vis diffuse reflectance spectra, however, revealed clear distinctions between the bare ZnO and its Ag decorated counterpart. As shown in Fig. 2b, the bare ZnO nanopowder exhibits intense absorption in the ultraviolet region ($<400 nm$) which is consistent with the wide band gap nature of the ZnO semiconductor. By strong contrast, its silver decorated

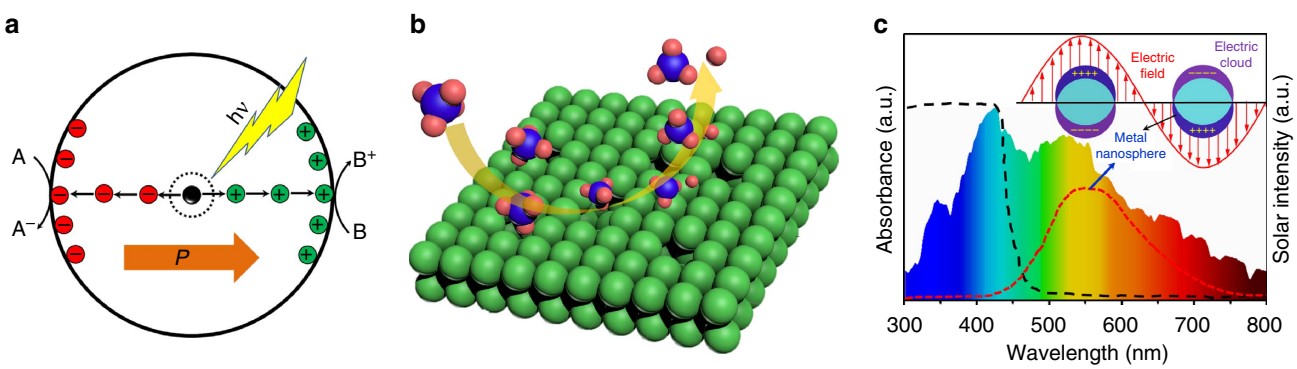

**Figure 1 | Materials design considerations.** (**a**) Polar structures favour fast separation and transportation of photo-generated electrons and holes. (**b**) Rich defective surfaces favour surface reactions. (**c**) Decorated metallic nanostructures may act as both a co-catalyst and a light-harvesting medium.

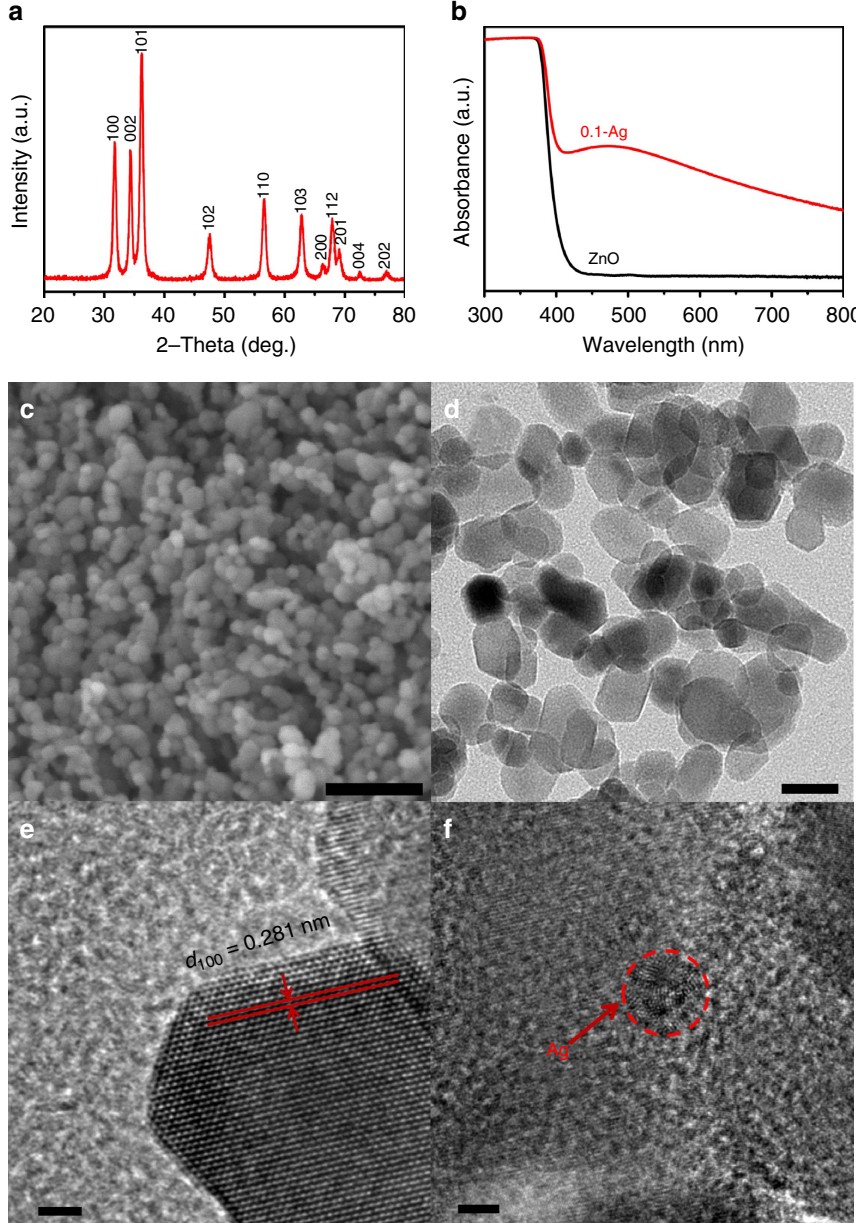

**Figure 2 | Physical characterization of the catalysts.** (**a**) Room temperature XRD patterns of the 0.1 wt% Ag decorated ZnO (0.1-Ag) powders. (**b**) Ultraviolet–visible diffusive reflectance spectra of the ZnO with and without Ag decoration. (**c**) SEM image of the 0.1-Ag powders. (**d**) TEM image of the 0.1-Ag powders. (**e,f**) HRTEM images of the 0.1-Ag sample. Scale bars, 100 nm (**c**), 20 nm (**d**) and 2 nm (**e,f**).

counterpart exhibits not only the intense ultraviolet absorption expected for the bare ZnO, but also a broad absorption in the visible light region (peaking at ~470 nm and extending to over 800 nm), owing to the strong surface plasmon resonance of the metallic Ag nanoparticles[31].

Morphologies of the samples were characterized by both scanning electron microscopy and transmission electron microscopy (TEM). The ZnO powder shows an irregular morphology with an average particle size of ~20 nm (Fig. 2c,d). High-resolution TEM observation further confirmed the crystal structure of ZnO where the interplanar lattice spacing of 0.281 nm corresponds well to the (100) plane of hexagonal wurtzite structure type of ZnO (Fig. 2e). Moreover, the high-resolution TEM analysis identified the particle size of silver that decorated on ZnO is only ~2 nm (Fig. 2f). Elemental mapping was further carried out to examine distribution of the silver nanoparticles and no obvious aggregation was detected.

**Photocatalytic properties characterization.** Photocatalytic $CH_4$ oxidation of the as-fabricated samples were examined under simulated sunlight illumination (see Supplementary Fig. 1) with both fixed-bed and flow-bed mode (see Supplementary Fig. 2). Figure 3a shows a typical time evolution of the methane photo-oxidation over the ZnO samples under the fixed-bed mode. For comparison purposes, the performance of commercial ZnO (see Supplementary Fig. 3: 200–300 μm particles size with the BET surface area of ~3.5 m$^2$ g$^{-1}$) and P25 (a recognized benchmark photocatalyst with the BET surface area of ~50 m$^2$ g$^{-1}$), under the same experimental conditions are also shown. It was found that ZnO possesses an obvious size effect on photocatalytic methane oxidation (see Supplementary Fig. 4), and, the nano-particulate ZnO exhibits exceptional activity for $CH_4$ oxidation either under ultraviolet or UV–vis light illumination. Ag decoration further enhances the photo-oxidation activity. By strong contrast, the commercial ZnO and P25 exhibit

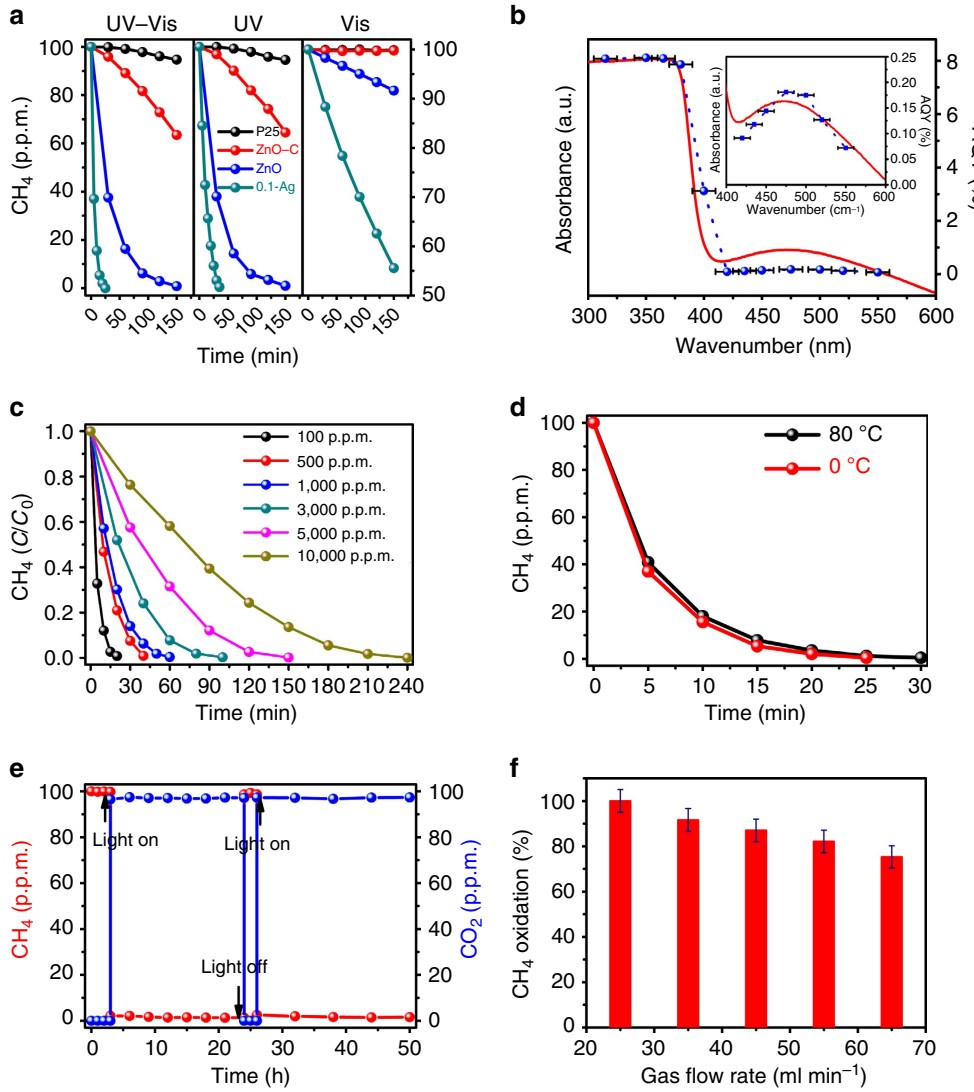

**Figure 3 | Photocatalytic oxidation of methane.** (**a**) Photocatalytic oxidation of methane in a fixed-bed mode with full arc (UV–vis), ultraviolet and visible light illumination, respectively. For comparison purposes, photo-activities of the commercial TiO$_2$ (P25), commercial ZnO (ZnO-C) and as-fabricated ZnO under the same experimental conditions were shown as well. (**b**) Ultraviolet–visible diffuse reflectance spectrum and AQYs of the 0.1-Ag sample plotted as a function of wavelength of the incident light. AQYs were plotted at the centre wavelengths of the band-pass filters, with error bars showing the deviation of the centre wavelengths ($\Delta\lambda = \pm 12$ nm). (**c**) Time evolution of the methane photo-oxidation over the 0.1-Ag sample in the fixed-bed mode under full arc illumination with various initial CH$_4$ concentration. (**d**) Influence of the temperature on the methane photo-oxidation activities over the 0.1-Ag sample under full arc illumination. (**e**) Methane photo-oxidation activity over the 0.1-Ag sample under full arc illumination and a flow-gas mode with gas flow rate of 25 ml min$^{-1}$. (**f**) Influence of the gas flow rate on the rate of methane oxidation under the flow-gas mode with $\pm 5\%$ error bars calculated from the sample introduction uncertainty.

only mild and faint activity, respectively, under the same illumination conditions. When illuminated under visible light, neither commercial ZnO nor P25 exhibit any activity for CH$_4$ oxidation, however, the nano-particulate ZnO still shows significant activity and the silver surface plasmon resonance enhancing methane photo-oxidation is undoubtedly corroborated herein.

The wavelength dependence of the CH$_4$ oxidation was then further investigated to prove whether or not the reaction really was driven by light. Figure 3b shows the UV–vis diffuse reflectance spectrum of the 0.1 wt% Ag decorated ZnO along with the apparent quantum yield (AQY) of methane oxidation as a function of the incident light wavelength. The AQY decreased with increasing wavelength in the ultraviolet region and the AQY in the visible light region was found to coincide with the characteristic absorption of the silver surface plasmon resonance.

This indicates that the methane oxidation reaction is indeed driven by light and that the light-absorption property of the Ag decorated ZnO semiconductor governs the reaction rate. The high quantum yield of 8% at wavelengths < 400 nm and over 0.1% at wavelengths ~470 nm, shows great promise for atmospheric methane oxidation.

In consideration of the knowledge that methane oxidation is an exothermal reaction[4,12,17], further experiments such as methane photo-oxidation under various initial hydrocarbon concentrations (Fig. 3c) and under different temperatures (Fig. 3d), were also carried out and the results indicate that temperature fluctuation has little effect on the photo-oxidation process. Careful analysis of the methane photo-oxidation (see Supplementary Fig. 5) revealed that the reactions follow pseudo-first-order kinetics and the apparent reaction rate constant $k$ deduced from the Langmuir–Hinshelwood model[34] decrease from 0.24 to 0.02 min$^{-1}$

when the initial methane concentration increase from 100 to 10,000 p.p.m. These results indicate that by strong contrast to thermal catalysis the approach of photocatalysis is much more promising for the elimination of low concentrations of methane that are difficult to cope with using thermal catalysis.

To examine the mineralization rate and also the carbon balance, the flow mode test was performed as well. Before illumination, $CO_2$ in the reaction system was expelled by flowing carrier gas. After that, the reaction gas consisting of 78.9% $N_2$, 21.1% $O_2$ and 100 p.p.m. methane was flowed through the Ag–ZnO samples and analysed directly by gas chromatography (GC9720 Fuli). During the reaction, a 300 W Xe lamp was used to provide simulated solar light with light density of $\sim$200 mW cm$^{-2}$. Figure 3e shows the time dependency of the $CH_4$ photo-oxidation on the Ag decorated ZnO catalysts under simulated sunlight illumination in the flow mode experiment. Before light was turned on, the detected concentration of $CH_4$ was 100 p.p.m. and no $CO_2$ was detected. When the lamp was turned on, the amount of methane decreased rapidly to $\sim$1.5 p.p.m. Simultaneously, the concentration of $CO_2$ increased promptly to $\sim$97.3 p.p.m. During the methane photo-oxidation reactions, no CO or other hydrocarbons were detected by gas chromatography. Carbon mass balance of 98.8% is thus obtained based on the ratio of carbon output (1.5 p.p.m. $CH_4$ and 97.3 p.p.m. $CO_2$) to carbon input (100 p.p.m. $CH_4$), which is close to 100% if the experimental uncertainty is considered. When the light was turned off, the concentration of $CO_2$ rapidly decreased to zero, and in the meantime, the amount of methane returned to the constant value. By contrast, the same experiment with thermal catalysis was performed as well. It was found that there is totally no activity of methane oxidation even heating the samples to 250 °C and decreasing the gas flow rate to 10 ml min$^{-1}$ (see Supplementary Fig. 6). The results again confirm that the methane oxidation occurs through a photo-driven process. Furthermore, the activities of the sample shown in Fig. 3e exhibit no decrease in the 50 h' flow-gas mode experiment, which evidence the high stability of the silver decorated ZnO catalysts.

The influence of gas flow rate on the oxidation of methane was also investigated (Fig. 3f). It was found that increasing the gas flow rate from 25 to 65 ml min$^{-1}$ caused the ratio of methane oxidation to decrease linearly from almost 100 to $\sim$76%, which is consistent with the fact that the photocatalytic reaction is a rate-determined process[35].

The turnover number (TON) of the $CH_4$ photo-oxidation was obtained by oxidizing a larger amount of $CH_4$ gases over the Ag decorated ZnO catalysts. It has been shown the methane oxidation is a photo-driven process. However, there is no activity if illuminating methane without the presence of the catalyst (see Supplementary Fig. 4c), the fact that the calculated TON for the $CH_4$ photo-oxidization is obviously greater than one (see Supplementary Note 1) indicates that the photo-oxidation reaction is truly driven by a catalytic process.

Photo-oxidation of other hydrocarbons such as ethane, propane, and ethylene were also carried out to further confirm the strong photo-oxidative ability of the silver decorated ZnO catalyst. Similar to methane, these small molecular hydrocarbon gases are difficult to oxidise under mild conditions because of their high bond energy as well as weak molecular polarity[22]. The highly efficient photo-activity for multiple hydrocarbon gases (see Supplementary Fig. 7) demonstrates that the silver decorated ZnO is a promising candidate for the treatment of atmospheric hydrocarbons under mild conditions.

Stability of a photocatalyst is one of the most important parameters for practical applications. A cycling $CH_4$ photo-oxidation test (see Supplementary Fig. 8) was thus

performed for this purpose. After ten cycles, the activity of the silver decorated ZnO semiconductors remains unchanged. After the aforementioned experiments, the Ag–ZnO samples were also carefully examined by XRD, optical absorption and X-ray photoelectron spectroscopy analysis. There are no noticeable distinctions between the freshly prepared and the repeatedly used samples (see Supplementary Fig. 9). These results indicate that the Ag–ZnO catalysts are indeed very stable for hydrocarbon photocatalytic oxidation.

**Photocatalytic *in situ* characterization.** To obtain further insight into the high photo-oxidative activity of the Ag decorated ZnO, *in situ* electron paramagnetic resonance (EPR) as well as Fourier transform infrared spectroscopy (FT-IR) studies have also been carried out. Figure 4a shows the EPR spectra collected on the Ag–ZnO sample under various atmospheres and illumination conditions. Under the dark and air atmosphere, the sample shows two signals with $g = 2.005$ and $g = 1.960$. The signal of $g = 2.005$ is assigned to single-electron-trapped surface defects such as $V_o^+$ or $O_s^-$ (refs 36,37), which is an important feature that is observed only when the particle size of ZnO decreases to the

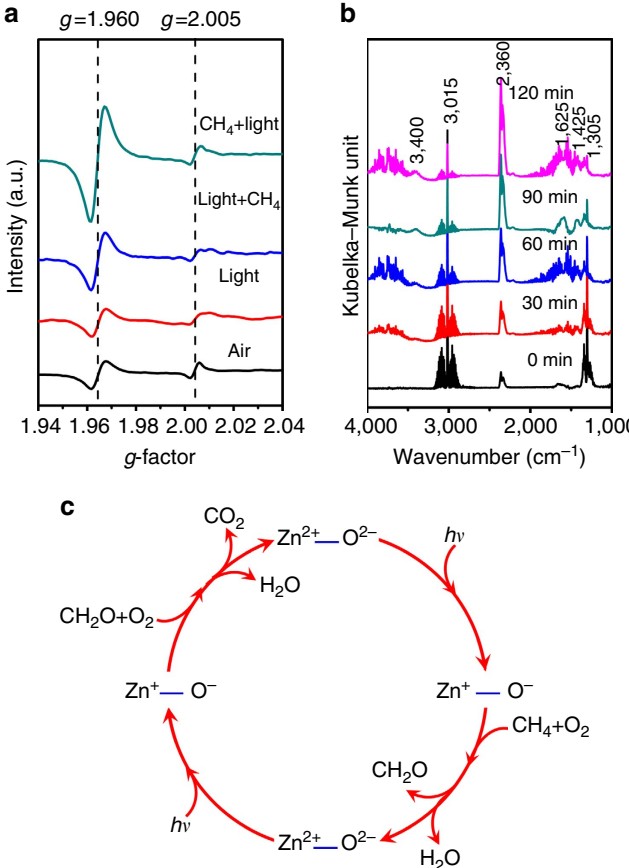

**Figure 4 | Mechanism of photocatalytic CH$_4$ oxidation.** (**a**) EPR signals of 0.1-Ag under different environments. From the bottom-up, the traces are for a fresh sample measured in an air atmosphere, measured in an air atmosphere after illumination, measured immediately after injection CH$_4$ to the illuminated system, measured after illumination under CH$_4$ and air atmosphere, respectively. (**b**) *In situ* IR spectra of methane photocatalytic oxidation collected at different illumination time intervals. (**c**) Schematic illustration for the photocatalytic CH$_4$ reaction processes under ambient conditions.

nanoscale. The signal with $g = 1.960$ is attributable to the lattice electron trapping sites ($Zn^+$ or $V_{Zn}^-$)[36,37] in the defect-rich semiconductor of ZnO. The intensity ratio of the two signals shows less change when illuminating the sample under air atmosphere. However, once methane is injected into the reactor, the signal of $Zn^+$ increases promptly while the signal of the single electron surface defects remains unchanged. For the sample in the atmosphere containing methane and oxygen, continuous illumination caused the signal of $Zn^+$ to keep increasing whereas the signal of surface defects ($V_o^+$ or $O_s^-$) increased only slightly (see Supplementary Fig. 10). In view of the fact that single electron defects $Zn^+$ and $O^-$ are always generated in pairs when illuminating ZnO, the changes of the EPR signals indicate that the surface defects ($V_o^+$ or $O_s^-$) play a vital role in the methane photo-oxidation.

Figure 4b shows *in situ* diffusive reflectance infrared spectra that was collected during the photocatalytic oxidation of methane. Methane is featured with typical IR vibration modes at $\sim 1,305$ and $\sim 3,015\,cm^{-1}$ as well as the multiple IR bands close to $3,015\,cm^{-1}$ (ref. 38). The IR bands at $\sim 2,340–2,360\,cm^{-1}$ are assigned to the characteristic mode of $CO_2$ (ref. 39). With light illumination, the decrease of the intensities of the bands assigned to the $\nu$(C-H) vibration of methane is accompanied by gradual increase of the intensities of the IR bands of $CO_2$. Meanwhile, the newly emerged broad peaks at $\sim 1,625$ and $\sim 3,400\,cm^{-1}$ keep rising, which correspond to $\delta$ (HOH) and $\nu$ (HOH) vibrations of chemisorbed $H_2O$[40], respectively. Significantly, the newly emerged band at $\sim 1,425\,cm^{-1}$ that corresponds to the $\delta$ (CHO) mode of chemisorbed aldehyde[41], shows less increase with the proceeding of light illumination. During the experiment, no other intermediate species was detected. These results revealed that the methane photo-oxidation, in all likelihood, proceeds via a two-step process (Fig. 4c): first, $CH_4$ reacts with $O_2$ and produces $H_2O$ and HCHO ($CH_4 + O_2 \rightarrow HCHO + H_2O$), and then the intermediate product HCHO further reacts with $O_2$ and produces $H_2O$ and $CO_2$ ($HCHO + O_2 \rightarrow CO_2 + H_2O$).

## Discussion

As we known, the primary step of methane activation on oxide materials frequently involves reaction with surface $O^-$ radical ions[42–44]:

$$oxide{-}O^\bullet + CH_4 \rightarrow {}^\bullet CH_3 + HO{-}oxide. \qquad (1)$$

When ZnO was illuminated under simulated solar light, surface electron ($Zn^+$) and hole ($O^-$) centres will generate via the reaction[45]:

$$Zn^{2+}{-}O^{2-} \xrightarrow{h\nu} Zn^+{-}O^-. \qquad (2)$$

Earlier research has demonstrated that the $Zn^+$ cations can attract three hydrogen atoms of methane and the fourth hydrogen is on the opposite side[27], whereas the $O^-$ anion has a strongly attractive force for the hydrogen atoms of methane and can abstract the fourth hydrogen from methane[36]. Therefore the surface-adsorbed $CH_4$ would be activated which will initiate the following reactions:

$$Zn^+{-}O^- \xrightarrow{CH_4} Zn^+ {}^\bullet CH_3{-}OH^-$$
$$\rightarrow Zn^{2+}{-}O^{2-} + {}^\bullet CH_3 + {}^\bullet H, \qquad (3)$$

$$O^- (hole) + OH^- \rightarrow O^{2-} + OH^\bullet, \qquad (4)$$

$$CH_4 + OH^\bullet \rightarrow {}^\bullet CH_3 + H_2O. \qquad (5)$$

Since oxygen was present in the reactor, the surface electron ($Zn^+$) would either get recombined with hole ($O^-$) to form

$Zn^{2+}$ and $O^{2-}$ or react with surface-adsorbed oxygen molecule to form $Zn^{2+}$ and adsorbed superoxide anion radicals:

$$Zn^+ (electron) + O_2 \rightarrow Zn^{2+} + O_2^-. \qquad (6)$$

The generation of superoxide anion radicals will initiate further oxidation of the methyl radicals:

$$^\bullet CH_3 + O_2^- \rightarrow CH_2O + OH^-. \qquad (7)$$

Since the superoxide anion radicals react very easily with the surface $OH^-$ to form their conjugated acid[46,47], the following route to generate formaldehyde cannot be ruled out:

$$OH^- (hole) + O_2^- \rightarrow O^{2-} + {}^\bullet O_2H, \qquad (8)$$

$$^\bullet CH_3 + {}^\bullet O_2H \rightarrow CH_3OOH \rightarrow HCHO + H_2O. \qquad (9)$$

We know the oxidation of formaldehyde has been extensively investigated. With the involvement of active oxygen species $O_2^-$, $^\bullet OH$ and $O^-$, the intermediate product formaldehyde can conveniently be oxidized to $CO_2$ and $H_2O$ in a similar manner[48,49].

The aforementioned analysis distinguishes photocatalytic methane oxidation from the thermocatalytic approach, where the latter requires the thermal activation of oxygen to drive the methane oxidation. This process is temperature dependent. Since $CH_4$ oxidation is an exothermic reaction, a higher concentration of methane releases more heat, which is beneficial for the activation of oxygen. Therefore, the thermocatalytic approach is more efficient for the treatment of methane if it is in high concentration. Whereas for the photocatalytic methane oxidation, the lattice oxygen activated by photo-generated hole is the main active species for abstracting the hydrogen of $CH_4$. This process is not determined by the reaction temperature but closely related to the light energy and intensity. Therefore, the photocatalytic oxidation is less sensitive to temperature fluctuations. Instead, once the illumination condition is fixed, the reaction rate will depend on the concentration of methane, and proceed more quickly for lower concentrations.

The function of nano silver decoration lies at least in: (I) as electron sink reducing the recombination of electrons and holes in the surface of ZnO (see the significantly reduced photo-luminescence spectra intensity in Supplementary Fig. 11); (2) as a photo-sensitizer extending the utilization of the visible light.

On the basis of the above understanding, one could predict that if no oxygen is involved in the methane photo-oxidation, ethane will be produced owing to the oxidative dehydrogenation of methane, and, if ethane further abstracts hydrogen the generation of ethylene and other hydrocarbons will occur. We then further performed the flow mode methane conversion experiments under oxygen-free conditions and a methane conversion of 0.35% and a selectivity of 89.47% for ethane were obtained (see Supplementary Fig. 12).

## Methods

**Sample preparation.** The nano-particulate ZnO powders were prepared by a method of precipitation: 0.005 mol $Zn(NO_3)_2$ and 0.005 mol oxalic acid were dissolved, respectively, in 100 ml distilled water at room temperature. Then, the oxalic acid solution was added into the $Zn(NO_3)_2$ solution drop by drop to get zinc oxalate precipitates. After that, the precipitates were filtered and calcined at 350 °C in air atmosphere for 6 h. The Ag–ZnO composite photocatalysts were prepared as follows: First, 1.00 g ZnO powers were dispersed into 100 ml aqueous solution that containing various amount of $AgNO_3$ in a quartz reactor under vigorous stirring. Then, the suspension was evaporated at 80 °C until dryness. After that, the precipitates were treated at 350 °C in air atmosphere for 2 h. For simplicity, the resultant Ag–ZnO composites with 0.1 wt% Ag (compared with ZnO) deposition were denoted as 0.1-Ag.

**Physical characterization.** The structure and crystallinity of the samples were investigated by XRD (Rigaku Miniflex II) using Cu $K_\alpha$ ($\lambda = 0.15418$ nm) radiation (30 kV, 15 mA). A scan rate of $5^o\,min^{-1}$ was applied to record the powder XRD

patterns in the 2θ range of 20–80°. The diffuse reflectance UV-visible spectra of the samples were recorded on a PerkinElmer Lambda 900 UV/VIS/NIR spectrometer that was equipped with an integrating sphere covered with $BaSO_4$ as the reference. The BET-specific surface areas of the samples were measured by a TriStar II 3020-BET/BJH Surface Area analyzer. Images of TEM and high-resolution TEM as well as electron diffraction patterns were obtained using a JEM 2010 EX instrument at an accelerating voltage of 200 kV. The X-ray photoelectron spectroscopy measurements were performed on a Phi Quantum 2000 spectrophotometer with Al $K_\alpha$ radiation (1,486.6 eV). The binding energies were calibrated using that of C 1 s (284.8 eV). Photoluminescence spectra of the photocatalysts were collected on a Varian Cary Eclipse spectrometer with an excitation wavelength of 325 nm. *In situ* FT-IR studies were performed on a spectrometer Nexus FT-IR (Thermo Nicolet) by using a diffuse reflectance attachment equipped with a reaction chamber. The 128 single-beam spectra had been co-added at a resolution of $4 \, cm^{-1}$ and the spectra were presented as Kubelka–Munk function referred to adequate background spectra. The background and samples spectra were taken (the average of accumulated 32 scans) over the frequency range $4,000–600 \, cm^{-1}$. The EPR spectra were obtained on a Brucker A300 spectrometer. The details of the instrumental parameters are as follows: scanning frequency: 9.85 GHz, central field: 3350 G, scanning width: 1,260 G, scanning power: 20 mW, and scanning temperature: 25 °C.

**Photocatalytic experiments.** The photocatalytic oxidation of hydrocarbons were carried out in a homemade fixed-bed pyrex reactor of 450 ml capacity (see Supplementary Fig. 2a) and a homemade flow-bed pyrex reactor of 0.6 ml $(30 \times 20 \times 1 \, mm^3)$ capacity (see Supplementary Fig. 2b), respectively. All of the experiments were performed at atmospheric pressure and room temperature unless otherwise stated. In a typical fixed-bed reaction: First, 0.5 g photocatalysts were dispersed uniformly on the bottom of reactor. Then, the reactor was flushed with 78.9% $N_2$ and 21.1% $O_2$ mix gas repeatedly to remove water and $CO_2$ that adsorbed on the catalyst and the inwall of reactor. Subsequently, different amounts of hydrocarbons were injected into the reactor by a micro-syringe. Before the illumination, the reactor was kept in the dark for 2 h to ensure the establishment of an adsorption-desorption equilibrium between the photocatalyst and reactants. Then, the reactor was illuminated by a 300 W Xe lamp from the upper part with light intensity of $\sim 200 \, mW \, cm^{-2}$. At a certain time interval, 4 ml gas was sampled from the reactor and analysed by a gas chromatograph (GC9720 Fuli) equipped with a HP-Plot/U capillary column, a molecular sieve $13 \times$ column, a flame ionization detector and a thermal conductivity detector. A typical flow-bed reaction proceeded as follows: first, 0.5 g photocatalysts were fully filled in the flow-bed pyrex reactor; second, the mixed gas consisting of 78.9% $N_2$, 21.1% $O_2$ and 100 p.p.m. hydrocarbons was flowed through the samples and analysed directly by the gas chromatograph (GC9720 Fuli). The reactor was illuminated using 300 W Xe lamp from both the top and bottom surfaces during the photoreactions. The oxygen-free conversion of methane was carried out using the same procedure and the only difference was the reaction gas which consisted of 95% $N_2$ and 5% $CH_4$ that free of oxygen.

The AQY measurements were performed with the fixed-bed mode and monochromatic light illumination for 2 h under different wavelength was used during the experiment. On the basis of the reaction $CH_4 + 2O_2 \rightarrow CO_2 + 2H_2O$ and the assumption that all electrons are excited by light, the AQYs are calculated by the following formula:

AQYs (%) = $100 \times$ (the number of reacted electrons or holes)/(the number of incident photons) = $100 \times$ (the number of reacted $CH_4$ molecules $\times 8$)/(the number of incident photons).

**Data availability.** The data that support the findings of this study are available from the corresponding author on request.

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

## Acknowledgements

This work was financially supported by the National Key Project on Basic Research (Grant No. 2013CB933203), the Strategic Priority Research Program of the Chinese Academy of Sciences (Grant No. XDB20000000), the Natural Science Foundation of China (Grant No. 21373224, 21577143 and 51502289), the Natural Science Foundation of Fujian Province (Grant No. 2014H0054 and 2015J05044) and the One Hundred Talents Program of the Chinese Academy of Sciences.

## Author contributions

X.C. prepared the samples and carried out the experiments; Y.L. assisted the photocatalytic tests; X.P. directed the IR analysis; D.C. contributed the manuscript revision; X.H. and Z.Y. co-supervised the project; X.C. and Z.Y. wrote the paper and all authors discussed the results and commented on the manuscript.

## Additional information

**Competing financial interests:** The authors declare no competing financial interest.

