## [Peer Review File · Nature Communications]

Reviewers' comments:

Reviewer #1 (Remarks to the Author):

In this submission, Chen et al described their research on photocatalytic oxidation of CH₄ by using a highly active Ag/ZnO catalyst. This work may draw some attentions in the fields of materials and catalysis, but I cannot recommend the publication on Nature Communications due to the following concerns.

- The authors claimed that ZnO fulfills several considerations for materials design, however their interpretation is not convincing because many types of photo-active oxides can satisfy these requirements. Also, size effect of ZnO was emphasized but only one kind of nano-ZnO was used, which cannot provide a full view of the proposed concept.
- Length-width ratio of TEM/SEM images cannot be definitely changed! If it was changed, the scale of one specific dimension is incorrect. Moreover, the content of Ag is as low as 0.1wt% in the sample, so the signal of Ag in elemental maps is probably coming from the background noise. It is not convincing to prove the uniform distribution of Ag.
- Carbon balance is of great importance to evaluate a CH₄ oxidation process, but the data is missed. The expressions of activity like CH₄(C/C₀), CH₄ (ppm) and CH₄ conversion are somehow disordered.
- The proposed reaction mechanism in fig. 4c is beyond of my imagination. If Zn(2+)-O(2-) was supposed as the adsorption sites for both CH₄ and its derivate HCHO, there would be highly competitive adsorption of one major reactant and its degraded intermediate. Meanwhile, Zn(+)-O(-) will anticipate the dehydrogenation of the adsorbed CH₄ and HCHO -- this mechanism is overcomplicated. The characterization actually cannot support this hypothesis. For instance, vibration of CHO at 1425 cm⁻¹ became more intensive as the reaction time increased until 120 min, but the reaction already ended for a while (fig. 3).
- The English should be thoroughly polished.

Reviewer #2 (Remarks to the Author):

This paper reports the results of the author's laboratory concerning the synthesis of ZnO and ZnO/Ag photocatalyst particles exhibiting an unusually high photocatalytic activity for the oxidation of methane in the gas phase even upon visible light illumination. The experimental work was obviously conducted with care, the results are being compared with the activity of two reference photo catalysts, namely, commercial ZnO and P25 TiO₂. The discussion presents a (rather speculative) reaction mechanism to explain the results. Since the photocatalytic oxidation of methane still presents considerable problems and the system presented here operates with quantum yields approaching 8 % it is suggested to accept this work for publication. Improvement of the usage of the English language is, however, highly indicated.

Reviewer #3 (Remarks to the Author):

The manuscript describes the photocatalytic oxidation of methane over a Ag-decorated ZnO catalyst using UV, UV-vis and visible radiation. The activity of the catalyst is dependent on the particle size of the ZnO with the activity of the nano-sized material prepared here significantly higher than the commercially available material. The presence of Ag is reported to act as an electron sink and also extend the region of use into the visible region which is shown by UV/DRS. The use of visible light for photocatalysis is of interest and its application for methane oxidation using visible light is also topical and challenging.

The paper presents a wide range of kinetic data and characterisation techniques which support the conclusions of the paper. The data is consistent, of good quality and where appropriate errors have been discussed. Although the error/reproducibility of the kinetic tests is not explicitly described it is inferred in the 10 runs on the same catalyst with no loss in activity and the continuous flow

experiments.

Although the data presented is consistent with the conclusions some additional information could further improve the manuscript and this is outlined below.

Suggestions:

The paper would need a thorough proof read before publication to correct spelling, grammatical errors and to improve phrasing.

As the manuscript should be the length of a communication, it is necessary to make the best use of the space available and the introduction could be rewritten to allow more emphasis on the catalyst design, the catalysts used for methane photocatalytic oxidation and use of ZnO as a photocatalyst rather than the detail given on page 3 on what photocatalysts need to do. It is not certain that Fig 1 is needed in the work. The novelty of the work would then be clearer.

If the experimental must be at the end of the paper then it would perhaps be clearer if there were no experimental details in the Results section of the paper on page 4. Also on page 9 the amount of methane used should be given rather than "a small quantity of methane was added". 100ppm is given later but it would be good to use this value at the start of the description of the experiment. A number of Ag loadings are given in the experimental but I think that only the results for the 0.1% are shown. If this is the case then the other loadings should be removed from the experimental section.

Page 4 says that no Ag particles were observed on the xrd due to their low loading which is likely true however later on page 6 we are told that the particle size was only 2nm which is well below the detection limit of the xrd. It would be good to link these two results as one supports the other.

On page 6, it is difficult to see how the cells presented in the ESI are operated as the experimental details are at the end of the paper. A reference to the experimental section in the ESI next to the figure would be helpful.

Figure 3 presents a lot of information and although it is clear enough the letters labelling the figures are rather large. Similarly for Fig 4.

Page 10 says that there was no decrease in conversion over 50 hours time on stream but I am not sure these results are shown.

Again as the paper needs to be kept condensed it would be useful to present the data in the most efficient way as possible and I wonder if this would give space to explain certain sections more clearly. In particular the section on using different concentrations of methane. This data could be used to determine the order of the reaction and would perhaps support the conclusion that photocatalysis is better than thermal methods for low concentrations of methane.

Also as there is no difference in rate on a change in temperature can this give us information on the difference in the thermal and photocatalytic mechanisms?

In fig 3f the x-axis is the gas flow rate which is described as the methane flow rate in the figure caption I am unsure that this is correct. There is no typical total flow rate (needed to calculate AQY in the experimental section so I may be wrong but all other methane concentrations are given in ppm.

Page 10 describes calculation of the TON which is fine but I am not sure about concluding that this is evidence of a catalytic reaction. If the reaction were solely happening in the presence of light (with no catalyst present) then the number of catalyst sites and number of molecules would not be related in a catalytic sense but the ratio of these individual values could easily be greater than 1. It would be good to add in the conversion in the absence of catalyst to confirm that this is a catalytic reaction. The reactions in Fig 3a indicate that it is catalysed ok but it would be better to confirm this with a blank reaction to draw the conclusions on page 10.

Improving the links between the different characterisation data and the kinetic data and clearly stating the key conclusions that can be drawn from each set of experiments presented would allow shortening of the paper and in turn give space for more detailed discussion.

There are a large number of references given from high impact journals and the abstract appropriately describes the work.

Response Letter

To Reviewer #1:

1. *In this submission, Chen et al described their research on photocatalytic oxidation of CH₄ by using a highly active Ag/ZnO catalyst. This work may draw some attentions in the fields of materials and catalysis, but I cannot recommend the publication on Nature Communications due to the following concerns.*

Thank you agreeing on this work may draw some attentions in the fields of materials and catalysis. Your concerns have been carefully considered and responded point by point shown in this letter.

2. *The authors claimed that ZnO fulfills several considerations for materials design, however their interpretation is not convincing because many types of photo-active oxides can satisfy these requirements. Also, size effect of ZnO was emphasized but only one kind of nano-ZnO was used, which cannot provide a full view of the proposed concept.*

Look back the history of photocatalysis we assume you acknowledge it is less possible to accurately predict a specific compound that can efficiently photo oxidize methane under mild conditions. The paragraph involving the materials design considerations is just used to address why we choose Ag-ZnO as our object of study. We never said only ZnO fulfills these requirements. In fact, a lot of semiconductors such as SrTiO₃, KNbO₃, CdS, Cu₂O, BiVO₄, g-C₃N₄ and Ag₃PO₄ etc. that have shown strong capabilities to drive water cleavage under light irradiation were carefully examined in our preliminary studies. None of the aforementioned semiconductors with known strong reduction or oxidation capabilities exhibits any activity for CH₄ photo-oxidation except P25 which shows a moderate photoactivity.

One paragraph is supplemented and the paragraph involving materials design consideration is revised as well (From Line 3-20, Page 3):

“In our preliminary studies, we fabricated a range of semiconductors including SrTiO₃, KNbO₃, CdS, Cu₂O, BiVO₄, g-C₃N₄ and Ag₃PO₄ etc. that have shown strong capabilities to drive water cleavage under light irradiation, using solid state reaction, hydrothermal, or other modified methods to examine their performance on driving methane photo-oxidation. None of

the aforementioned semiconductors, which are known to have strong reduction or oxidation capabilities, exhibit any activity for CH₄ photo-oxidation except P25 (a kind of nano TiO₂) which shows a moderate photoactivity. Heterojunction interface design²⁴, morphology control²⁵, and band edge modulation²⁶ were successively also employed to fabricate photoactive materials to address the photooxidation of small molecular hydrocarbons. Some small molecular hydrocarbons such as C₂H₆, C₃H₈ and C₂H₄ can be efficiently treated by these techniques, however, effective treatment of methane still remains a great challenge.”

“In light of the possibility that zinc ions may play an important role in methane activation²⁷, we then turned to zinc containing compounds such as ZnO in order to examine its activity on photo-oxidizing methane. It should be noted that, although it has been extensively studied, ZnO has never been recognized as an efficient photocatalyst because of its limited light harvesting ability and serious photocorrosion problem.”

Size effect of ZnO was supplemented (From Line 13-14, Page 7).

“...It was found that, ZnO possesses an obvious size effect on photocatalytic methane oxidation (see Supplementary Fig.S4),...”.

Figure S4 Size effect of ZnO on photocatalytic methane oxidation. a, Room temperature XRD patterns of ZnO powders prepared at various calcination temperatures (°C). **b**, SEM images of ZnO powders prepared at various calcination temperatures (°C). The particles size are ~20, 25-40, 100-150, and 300-500 nm, respectively. **c**, Time course of methane photooxidation with and without the ZnO samples that prepared at various calcination temperatures (°C) Test mode: the fixed-bed with full arc illumination.

3. *Length-width ratio of TEM/SEM images cannot be definitely changed! If it was changed, the scale of one specific dimension is incorrect. Moreover, the content of Ag is as low as 0.1wt% in the sample, so the signal of Ag in elemental maps is probably coming from the background noise. It is not convincing to prove the uniform distribution of Ag.*

Figure 2 was updated where the errors have been corrected. Thank you! The sentence concerning elemental mapping was revised as follows (From Line 3-4, Page 7):

“...Elemental mapping was further carried out to examine distribution of the silver nanoparticles and no obvious aggregation was detected.”

4. *Carbon balance is of great importance to evaluate a CH₄ oxidation process, but the data is missed. The expressions of activity like CH₄(C/C₀), CH₄ (ppm) and CH₄ conversion are somehow disordered.*

The carbon balance was examined in the flow gas mode test (Figure 3e), and, the below revisions were made to emphasis this (From Line 17-20, Page 10):

“...During the methane photooxidation reactions, no CO or other hydrocarbons were detected by the gas chromatography. Carbon mass balance of 98.8% is thus obtained based on the ratio of carbon output (1.5 ppm CH₄ and 97.3 ppm CO₂) to carbon input (100 ppm CH₄), which is close to 100%, if the experimental uncertainty is considered....”

The expressions of activity like CH₄(C/C₀), CH₄ (ppm) and CH₄ conversion are carefully checked and corrected in both Figure 3 and the manuscript text.

5. *The proposed reaction mechanism in fig. 4c is beyond of my imagination. If Zn²⁺-O²⁻ was supposed as the adsorption sites for both CH₄ and its derivate HCHO, there would be highly competitive adsorption of one major reactant and its degraded intermediate. Meanwhile, Zn⁺-O⁻ will anticipate the dehydrogenation of the adsorbed CH₄ and HCHO -- this mechanism is overcomplicated. The characterization actually cannot support this hypothesis. For instance, vibration of CHO at 1425 cm⁻¹ became more intensive as the reaction time increased until 120 min, but the reaction already ended for a while (fig. 3).*

Dear Reviewer, we carefully considered your comment and tried to simplify Figure 4c and

the mechanism discussion. Firstly, we have to make clear some fundamental facts: (1) The CH_4 is not depleted in 120 min of reaction (please note the IR vibration mode of methane at $\sim 3015\text{ cm}^{-1}$ in Figure 4b) since the in situ IR experiment was carried out with a large amount of methane to facilitate the detection of intermediate products. (2) The vibration of CHO at 1425 cm^{-1} shows less increase in comparison with the significantly rising of CO_2 . These experimental facts supported exactly the point that there is competitive adsorption between the reactant CH_4 and its degraded intermediate HCHO. The generated intermediate product HCHO will adsorb preferentially on the active sites and then experience further oxidation which prevented it from significant rising as that of CO_2 .

Since photocatalytic HCHO oxidation has been significantly investigated herein we omit its oxidation details and focus on the oxidation of methane to produce HCHO. Dehydrogenation of methane (reaction (1) in Discussion in Page 15) has been recognized in C1 chemistry as the primary step of methane activation. The reaction (2) is evidenced by the EPR results. The results of our flow mode methane conversion test under oxygen-free conditions provide a great support to the reactions (3-5) in Discussion in Page 15. With the detection of the intermediate HCHO, we trust you need not us to explain the following reactions (6-9) any further since they have been demonstrated in many classical literatures of photocatalysis [For example, *Chem. Rev.* 102, 3811-3836 (2002) and *Chem. Rev.* 95, 69-96 (1995)].

6. *The English should be thoroughly polished.*

Dr. David Cortie of the Australian National University kindly polished the English of our manuscript who is a native English speaker.

Thank you!

To Reviewer #2:

This paper reports the results of the author's laboratory concerning the synthesis of ZnO and ZnO/Ag photocatalyst particles exhibiting an unusually high photocatalytic activity for the oxidation of methane in the gas phase even upon visible light illumination. The experimental work was obviously conducted with care, the results are being compared with the activity of two reference photo catalysts, namely, commercial ZnO and P25 TiO_2 . The discussion

presents a (rather speculative) reaction mechanism to explain the results. Since the photocatalytic oxidation of methane still presents considerable problems and the system presented here operates with quantum yields approaching 8 % it is suggested to accept this work for publication. Improvement of the usage of the English language is, however, highly indicated.

Thank you for your positive response! The reaction mechanism was modified slightly to make it better supported and the English was corrected by David Cortie who is a native English speaker.

To Reviewer #3:

1. *The manuscript describes the photocatalytic oxidation of methane over a Ag-decorated ZnO catalyst using UV, UV-vis and visible radiation. The activity of the catalyst is dependent on the particle size of the ZnO with the activity of the nano-sized material prepared here significantly higher than the commercially available material. The presence of Ag is reported to act as an electron sink and also extend the region of use into the visible region which is shown by UV/DRS.*

The use of visible light for photocatalysis is of interest and its application for methane oxidation using visible light is also topical and challenging.

The paper presents a wide range of kinetic data and characterisation techniques which support the conclusions of the paper. The data is consistent, of good quality and where appropriate errors have been discussed. Although the error/reproducibility of the kinetic tests is not explicitly described it is inferred in the 10 runs on the same catalyst with no loss in activity and the continuous flow experiments.

Although the data presented is consistent with the conclusions some additional information could further improve the manuscript and this is outlined below.

Thank you for your positive comments and constructive suggestions!

2. *The paper would need a thorough proof read before publication to correct spelling,*

grammatical errors and to improve phrasing.

Done. Thank you!

3. *As the manuscript should be the length of a communication, it is necessary to make the best use of the space available and the introduction could be rewritten to allow more emphasis on the catalyst design, the catalysts used for methane photocatalytic oxidation and use of ZnO as a photocatalyst rather than the detail given on page 3 on what photocatalysts need to do. It is not certain that Fig 1 is needed in the work. The novelty of the work would then be clearer.*

The introduction was revised according to your suggestion and the below paragraphs were supplemented in the revised manuscript. We did not condense the paper as Nature Communications publishes papers of all lengths.

(From Line 3-20, Page 3):

“In our preliminary studies, we fabricated a range of semiconductors including SrTiO₃, KNbO₃, CdS, Cu₂O, BiVO₄, g-C₃N₄ and Ag₃PO₄ etc. that have shown strong capabilities to drive water cleavage under light irradiation, using solid state reaction, hydrothermal, or other modified methods to examine their performance on driving methane photo-oxidation. None of the aforementioned semiconductors, which are known to have strong reduction or oxidation capabilities, exhibit any activity for CH₄ photo-oxidation except P25 (a kind of nano TiO₂) which shows a moderate photoactivity. Heterojunction interface design²⁴, morphology control²⁵, and band edge modulation²⁶ were successively also employed to fabricate photoactive materials to address the photooxidation of small molecular hydrocarbons. Some small molecular hydrocarbons such as C₂H₆, C₃H₈ and C₂H₄ can be efficiently treated by these techniques, however, effective treatment of methane still remains a great challenge.”

“In light of the possibility that zinc ions may play an important role in methane activation²⁷, we then turned to zinc containing compounds such as ZnO in order to examine its activity on photo-oxidizing methane. It should be noted that, although it has been extensively studied, ZnO has never been recognized as an efficient photocatalyst because of its limited light harvesting ability and serious photocorrosion problem.”

Thank you!

4. *If the experimental must be at the end of the paper then it would perhaps be clearer if there were no experimental details in the Results section of the paper on page 4. Also on page 9 the amount of methane used should be given rather than "a small quantity of methane was added". 100ppm is given later but it would be good to use this value at the start of the description of the experiment.*

We revised the two parts as you suggested. Thanks!

5. *A number of Ag loadings are given in the experimental but I think that only the results for the 0.1% are shown. If this is the case then the other loadings should be removed from the experimental section.*

The other loadings were removed for the manuscript. Thank you!

6. *Page 4 says that no Ag particles were observed on the xrd due to their low loading which is likely true however later on page 6 we are told that the particle size was only 2nm which is well below the detection limit of the xrd. It would be good to link these two results as one supports the other.*

Done, thank you!

(From Line 7-9, Page 5) "...no diffraction peaks were detected for Ag owing to its low volume fraction (Figure 2a) and fine particle size (to be shown in Figure 2e)...."

7. *On page 6, it is difficult to see how the cells presented in the ESI are operated as the experimental details are at the end of the paper. A reference to the experimental section in the ESI next to the figure would be helpful.*

Done, thank you!

Figure S2 The schematic diagram of photocatalytic instruments. a, fixed-bed mode; b, flow-bed mode. For operation details please see the section of Photocatalytic experiments in Methods following the manuscript text."

8. *Figure 3 presents a lot of information and although it is clear enough the letters labelling the figures are rather large. Similarly for Fig 4.*

The figures are updated. Thank you!

9. *Page 10 says that there was no decrease in conversion over 50 hours time on stream but I am not sure these results are shown.*

This sentence was revised as follows (From Line 6-8, Page 11):

“...Furthermore, the activities of the sample shown in Figure 3e exhibit no decrease in the 50 hours experiment, which evidence the high stability of the silver decorated ZnO catalysts.”

10. *Again as the paper needs to be kept condensed it would be useful to present the data in the most efficient way as possible and I wonder if this would give space to explain certain sections more clearly. In particular the section on using different concentrations of methane. This data could be used to determine the order of the reaction and would perhaps support the conclusion that photocatalysis is better than thermal methods for low concentrations of methane.*

Thank you for your suggestion! The following sentences were supplemented (From Line 21, Page 9 to Line 2, Page 10):

“...Careful analysis on the methane photo-oxidation (see Supplementary Fig.S5) revealed that the reactions follow pseudo-first-order kinetics and the apparent reaction rate constant k deduced from the Langmuir–Hinshelwood model³⁴ decreases from 0.24 to 0.02 min⁻¹ when the initial methane concentration increases from 100 to 10000 ppm. These results indicate that...”

³⁴Hoffmann, M. R., Martin, S. T., Choi, W., Bahnemann, D.W. Environmental applications of semiconductor photocatalysis. *Chem. Rev.* **95**, 69-96 (1995).

Figure S5 Pseudo-first-order reaction kinetics plots of photocatalytic methane oxidation over the 0.1-Ag-ZnO samples with various CH₄ concentrations. The rate constants k deduced are 0.242, 0.112, 0.083, 0.051, 0.033 and 0.020 min⁻¹, respectively, with increasing the methane concentration from 100 to 10000 ppm.

11. Also as there is no difference in rate on a change in temperature can this give us information on the difference in the thermal and photocatalytic mechanisms?

One paragraph was supplemented to discuss the difference in the thermal and photocatalytic mechanism (From Line 14, Page 16 to Line 4, Page 17):

“The aforementioned analysis distinguishes photocatalytic methane oxidation from the thermocatalytic approach, where the latter requires the thermal activation of oxygen to drive the methane oxidation. This process is temperature dependent. Since CH₄ oxidation is an exothermic reaction, a higher concentration of methane releases more heat, which is beneficial for the activation of oxygen. Therefore, the thermocatalytic approach is more efficient for the treatment of methane if it is in high concentration. Whereas for the photocatalytic methane oxidation, the lattice oxygen activated by photo-generated hole is the main active species for

abstracting the hydrogen of CH₄. This process is not determined by the reaction temperature but closely related to the light energy and intensity. Therefore, the photocatalytic oxidation is less sensitive to temperature fluctuations. Instead, once the illumination condition is fixed, the reaction rate will depend on the concentration of methane, and proceed more quickly for lower concentrations.”

12. *In fig 3f the x-axis is the gas flow rate which is described as the methane flow rate in the figure caption I am unsure that this is correct. There is no typical total flow rate (needed to calculate AQY in the experimental section so I may be wrong but all other methane concentrations are given in ppm.*

The x-axis in Figure 3f was corrected as gas flow rate. The apparent quantum yield (AQY) measurements were conducted with the fixed-bed mode.

13. *Page 10 describes calculation of the TON which is fine but I am not sure about concluding that this is evidence of a catalytic reaction. If the reaction were solely happening in the presence of light (with no catalyst present) then the number of catalyst sites and number of molecules would not be related in a catalytic sense but the ratio of these individual values could easily be greater than 1. It would be good to add in the conversion in the absence of catalyst to confirm that this is a catalytic reaction. The reactions in Fig 3a indicate that it is catalysed ok but it would be better to confirm this with a blank reaction to draw the conclusions on page 10.*

The blank reaction was supplemented and the manuscript was revised accordingly (From Line 15-18, Page 11):

“...As there was totally no activity if illuminating methane without the presence of the catalyst (see Supplementary Figure S4c), the fact that the calculated TON for the CH₄ photo-oxidization was obviously larger than one ...”

Figure S4c Time course of methane photooxidation with and without the ZnO samples that prepared at various calcination temperatures (°C) Test mode: the fixed-bed with full arc illumination.

14. *Improving the links between the different characterisation data and the kinetic data and clearly stating the key conclusions that can be drawn from each set of experiments presented would allow shortening of the paper and in turn give space for more detailed discussion.*

We carefully made the revisions but did not condense the paper as Nature Communications publishes papers of all lengths. Thank you!

15. *There are a large number of references given from high impact journals and the abstract appropriately describes the work.*

Thank you for your positive comments!

REVIEWERS' COMMENTS:

Reviewer #1 (Remarks to the Author):

The authors have done quite a bit revision and all the questions have been appropriately addressed. I would like to recommend accepting the revised manuscript and publish it on Nature Communications.

Reviewer #3 (Remarks to the Author):

Manuscript

The manuscript is much better than the previously submitted version and the addition of the requested experiments and discussion is appreciated, however the standard of the work is still not at a level that could be published in Nature Communications as presented. Some suggestions to improve English spelling and grammar are given below.

In general there are too many commas and many are in the wrong places. There should not be a comma after an "and". It is only necessary to add a comma when separating one part of a sentence from another when there is a clear pause but in many cases here the comma is not needed. Some (but by no means all) corrections are given below.

12-13: to the nanoscale,

14: and, nano-Ag (no comma needed)

19: ethylene, and, in particular (no comma needed)

23: fuel and is an

25: life, the emissions

29: methane emissions is attracting

31-32: Over the course of a century, it has a greenhouse gas effect that is more than twenty times greater than the effect from the equivalent...

47: are the preliminary results presented in a previous publication? If so this should be referenced.

53: delete (a kind of nano TiO₂) but add in TiO₂ so it becomes "P25 TiO₂" as the reader should know what P25 is.

55: modulation were also successively employed (should this be successfully?)

79: should be "Ag decorated ZnO was chosen in this study not only because ZnO..."?

82: renders

96: Absorption in the UV region

97: of the ZnO semiconductor

99: absorption in the visible light

124: no comma needed after conditions. No comma needed after found that

126: exhibits

155: in the UV region. Also in the visible light region

166: has little effect. Analysis of the methane

169: decrease from

170: no comma needed after that at the end of the line

171: no comma needed after catalysis

178: by gas chromatography

186: by gas chromatography

191: the sentence should be reworded to remove the word we.

194-196: The results again confirm that the methane oxidation occurs through a phot-driven process. I am not sure what the next part of this sentence means so it should be reworded or removed.

196: sentence starting Furthermore the activities... should be rewritten to explain what is the 50 hr experiment.

205: over the Ag decorated ...

The sentence ending on 209 needs to be rewritten to clarify the conclusions: The fact that there is no activity in the absence of illumination indicates that this is a light driven process and this has already been discussed. Given that this is the case, the fact that the TON is greater than 1 indicates that this is a catalytic process.

213: are difficult to oxidise

226: stable for hydrocarbon photocatalytic oxidation

238: in an air atmosphere, measured in an air atmosphere

244: decreases to the nanoscale.

263 I am not sure distinctly is the correct word here. Perhaps, significantly would be better?

264 corresponds to

267: proceeds

280: strongly attractive force

309-313: these conclusions are corroborated by the results presented in the manuscript and this should be mentioned

318-324: a number of grammatical errors in this paragraph

326: particulate ZnO powders

Experimental contains a number of errors and should be thoroughly proof read again.

ESI

30: ZnO that was decoration with

34: and ZnO

40: the particles sizes are

42: that were prepared

86; and its decorated counterpart.

91: that was carried out

92: the original spectrum

104: gained and lost

106: mol should either all be in italics or not but not one of each

107: electron's gain and loss in the photocatalytic degradation

Response Letter

To Reviewer #1:

The authors have done quite a bit revision and all the questions have been appropriately addressed. I would like to recommend accepting the revised manuscript and publish it on Nature Communications.

Thank you!

To Reviewer #3:

Manuscript

The manuscript is much better than the previously submitted version and the addition of the requested experiments and discussion is appreciated, however the standard of the work is still not at a level that could be published in Nature Communications as presented. Some suggestions to improve English spelling and grammar are given below.

In general there are too many commas and many are in the wrong places. There should not be a comma after an "and". It is only necessary to add a comma when separating one part of a sentence from another when there is a clear pause but in many cases here the comma is not needed. Some (but by no means all) corrections are given below.

12-13: to the nanoscale,

14: and, nano-Ag (no comma needed)

19: ethylene, and, in particular (no comma needed)

23: fuel and is an

25: life, the emissions

29: methane emissions is attracting

31-32: Over the course of a century, it has a greenhouse gas effect that is more than twenty times greater than the effect from the equivalent...

47: are the preliminary results presented in a previous publication? If so this should be referenced.

53: delete (a kind of nano TiO₂) but add in TiO₂ so it becomes "P25 TiO₂" as the reader should

know what P25 is.

55: modulation were also successively employed (should this be successfully?)

79: should be "Ag decorated ZnO was chosen in this study not only because ZnO...?"

82: renders

96: Absorption in the UV region

97: of the ZnO semiconductor

99: absorption in the visible light

124: no comma needed after conditions. No comma needed after found that

126: exhibits

155: in the UV region. Also in the visible light region

166: has little effect. Analysis of the methane

169: decrease from

170: no comma needed after that at the end of the line

171: no comma needed after catalysis

178: by gas chromatography

186: by gas chromatography

191: the sentence should be reworded to remove the word we.

194-196: The results again confirm that the methane oxidation occurs through a phot-driven process. I am not sure what the next part of this sentence means so it should be reworded or removed.

196: sentence starting Furthermore the activities... should be rewritten to explain what is the 50 hr experiment.

205: over the Ag decorated ...

The sentence ending on 209 needs to be rewritten to clarify the conclusions: The fact that there is no activity in the absence of illumination indicates that this is a light driven process and this has already been discussed. Given that this is the case, the fact that the TON is greater than 1 indicates that this is a catalytic process.

213: are difficult to oxidise

226: stable for hydrocarbon photocatalytic oxidation

238: in an air atmosphere, measured in an air atmosphere

244: decreases to the nanoscale.

263: I am not sure distinctively is the correct word here. Perhaps, significantly would be better?

264: corresponds to

267: proceeds

280: strongly attractive force

309-313: these conclusions are corroborated by the results presented in the manuscript and this should be mentioned

318-324: a number of grammatical errors in this paragraph

326: particulate ZnO powders

Experimental contains a number of errors and should be thoroughly proof read again.

ESI

30: ZnO that was decoration with

34: and ZnO

40: the particles sizes are

42: that were prepared

86; and its decorated counterpart.

91: that was carried out

92: the original spectrum

104: gained and lost

106: mol should either all be in italics or not but not one of each

107: electron's gain and loss in the photocatalytic degradation

Thank you for your careful reading and very valuable suggestions. The language was carefully corrected and all the changes were shown in the revised manuscript text with track changes mode. The ESI is also carefully corrected point by point.

Thank you!